# Pharmacological treatment of depression: A systematic review comparing clinical practice guideline recommendations

Franciele Cordeiro Gabriel[1]*, Daniela Oliveira de Melo[2], Renério Fráguas[3], Nathália Celini Leite-Santos[1], Rafael Augusto Mantovani da Silva[2], Eliane Ribeiro[1]

1 Departamento de Farmácia, Faculdade de Ciências Farmacêuticas, Universidade de São Paulo, São Paulo, São Paulo, Brasil, 2 Departamento de Ciências Farmacêuticas, Instituto de Ciências Ambientais, Químicas e Farmacêuticas, Universidade Federal de São Paulo, Diadema, São Paulo, Brasil, 3 Departamento e Instituto de Psiquiatria, Hospital das Clínicas, Faculdade de Medicina da Universidade de São Paulo (IPq-HC-FM-USP); Divisão de Psiquiatria e Psicologia Hospital Universitário (HU) USP. Laboratório de Investigações Médicas – 21 (LIM 21) FM-USP, São Paulo, Brasil

* francielegabriel@usp.br

**Data Availability Statement:** All relevant data are within the manuscript and its Supporting Information files.

## Abstract

Depression affects over 300 million individuals worldwide and is responsible for most of the 800,000 annual suicides. Clinical practice guidelines (CPGs) for treatment of depression, founded on scientific evidence, are essential to improve patient care. However, economic and sociocultural factors may influence CPG elaboration, potentially leading to divergences in their recommendations. Consequently, we analyzed pharmacological recommendations for the treatment of depression from the most relevant CPGs. We included four CPGs with scores ≥ 80% for Domain 3 (rigor of development) on the Appraisal of Guidelines for Research and Evaluation and two other commonly used CPGs. The recommendations, their strengths, and the level of evidence were extracted from each CPG by two independent researchers and grouped as follows: (1) general recommendations for the pharmacological treatment for depression (suicide risk, acute treatment, continuation and maintenance phases, and treatment discontinuation); (2) treatment of non-responsive or partially responsive patients; and (3) treatment for subtypes of depression (chronic, psychotic, catatonic, melancholic, seasonal, somatic, mixed, and atypical). Only 50% of CPGs included recommendations for the risk of suicide associated with pharmacotherapy. All CPGs included serotonin selective reuptake inhibitors (SSRIs) as first-line treatment; however, one CPG also included agomelatine, milnacipran, and mianserin as first-line alternatives. Recommendations for depression subtypes (catatonic, atypical, melancholic) were included in three CPGs. The strength of recommendation and level of evidence clearly differed among CPGs, especially regarding treatment augmentation strategies. We conclude that, although CPGs converged in some recommendations (e.g., SSRIs as first-line treatment), they diverged in cardinal topics including the absence of recommendations regarding the risk of suicide associated with pharmacotherapy. Consequently, the recommendations listed in a specific CPG should be followed with caution.

**Funding:** This study was financed by the Coordenação de Aperfeiçoamento de Pessoal de Nível Superior - Brasil (CAPES) - Finance Code 001 to FCG and NCL. The funder had no role in study design, data collection and analysis, decision to publish, or preparation of the manuscript. No additional external funding was provided for this study.

**Competing interests:** The authors have declared that no competing interests exist.

## Introduction

Globally, mental illness affects approximately 22% of the population [1]. Depression is the most prevalent psychiatric disorder, which affects more than 300 million individuals [2]. It is an incapacitating disorder, responsible for most of the 800,000 annual suicides [2]. Along with population growth and aging, the number of individuals with depression has also increased considerably and led to overloaded healthcare systems, thereby generating the need for resource optimization [1,3]. A primary challenge in the field of mental health is the development of health interventions based on scientific evidence to combat depression [4].

Carefully developed clinical practice guidelines (CPGs) can improve patient healthcare by outlining practices recommended based on scientific research [5–7].

CPGs should ensure that potential biases are properly approached during the development process and established recommendations have the viability to be implemented [8]. The development process of original high-quality CPGs demands time, resources, and an experienced team [9]. Scarcity of resources, particularly in developing countries, restricts the development of CPGs, potentially compromising their quality and validity [9]. Additionally, potential biases might result from cultural issues, even in developed countries. Recently, there has been an increase in the number of CPG publications, and problems concerning their quality have been highlighted in various studies [10–12].

The current study aimed to analyze the most relevant CPGs for the pharmacological treatment of depression and clarify a matrix of recommendations including agreements and potential disagreements among CPGs. Such a matrix can contribute to the development of a critical view of CPGs for practitioners and possibly help the development and adaptation of CPGs.

## Materials and methods

### Identification of clinical practice guidelines

We recently reported CPGs for the pharmacological treatment of noncommunicable diseases that could be considered "high-quality" [12]. In that study, we conducted individual systematic reviews for each included disease; and using the second version of the *Appraisal of Guidelines for Research and Evaluation (AGREE II)*, evaluated 421 CPGs to establish the quality of their protocol registered on PROSPERO (CRD42016043364) [12]. In this study, we focus specifically on the part of that systematic review about the pharmacological treatment of depression [12].

We conducted a comprehensive search in MEDLINE, Embase, and the Cochrane Library, as well as in 12 specific websites for CPGs, because all such databases are well-recognized guideline repositories that have been cited frequently in previous studies of systematic reviews [12, 13]. The CPG searched were published between 2011 and 2016 (details of search strategies are in S1 Appendix). In April 2019, we searched the literature to update the included CPGs. Two independent reviewers screened the records regarding the eligibility criteria and conducted the data extraction. Discrepancies were solved by consensus.

To be included in this study, a CPG for the treatment of depression should have recommendations concerning the pharmacological treatment of depression in adults. CPGs published in English, Portuguese, or Spanish with free or restricted access to updated versions were eligible. To be included in the analysis of recommendations the CPG should have been considered of high quality (see next paragraph or had to be one of the most well-known and widely accepted CPGs [14]. All CPGs included in the analysis of the recommendations was evaluated to verify the presence of their updated version until April 2019. CPGs were excluded if they were specific for inpatients; specific for local use; focused on specific treatment approaches, such as

psychotherapy or neuromodulation; or were for specific groups including pregnant women and children. CPGs addressing comorbidities in depression were also excluded.

CPG quality was judged by three independent appraisers on the basis of the six domains of the *AGREE-II*, as described previously [12]. Of these six domains, domain 3 (rigor of development) is considered the most relevant for the reliability of the recommendations[15–17]. The *AGREE II* does not suggest a cutoff value denoting acceptable or high quality; instead, cutoff values were determined by groups assessing CPG quality [17]. The domain 3 cut-off of ≥80% was adopted for this study to indicate high quality, as proposed in prior studies [18–20]. Details of quality appraisal are shown in S3 Appendix.

### Extraction and analysis of recommendations

All recommendations regarding pharmacological treatment and the classification of the level of evidence from the included CPGs (when this information was available) were independently extracted by two researchers. Disagreements between the researchers (FCG and NCLS) were resolved by consensus; in the absence of a consensus, a senior investigator (ER) was included to solve the disagreement.

To perform the analyses, recommendations were classified based on their type and organized into tables by main topics. One of the authors (FCG) developed the first version of the classification, which was discussed with professors of pharmacy (ER) and psychiatry (RF).

The final version of the comparative tables of recommendations were achieved after three rounds of discussion. The recommendations were grouped by the main topics: 1) general recommendations for pharmacological treatment of depression (acute suicide risk and treatment, continuation phase treatment, maintenance phase treatment, and treatment discontinuation), 2)recommendations for treatment for those who did not respond or partially responded to therapy, and 3) recommendations for the treatment of depression subtypes (chronic depression or dysthymia, psychotic depression, catatonic depression, atypical conditions, melancholic depression, seasonal depression, somatic depression, and mixed depression).

## Results

In our initial search, we identified 947 citations and abstracts after removing duplicates. Thereafter, by reading the full text and applying the eligibility criteria, we selected 27 CPGs for this study (Fig 1). (S2 Appendix includes the reason for excluding 105 full records).

For the analysis of recommendations, 6 CPGs were included. Four CPGs presented a score ≥ 80% for Domain 3 and were considered high-quality [21–24]. In addition to these selected CPGs, two others were included based on their widespread acceptance [14]: the Canadian Network for Mood and Anxiety Treatments (CANMAT) and the American Psychiatric Association (APA) guidelines [25,26]. The six CPGs selected for analysis of their recommendation, based on their AGREE II Domain 3 score or on their acceptability, were as follows: Guía Clínica AUGE [21], score = 89%; Guía de Práctica Clínica [22], score = 86%; Depression in adults [23], score = 84%; Depression, adult in primary care [24], score = 81%; Practice guideline for the treatment of patients with major depressive disorder [26], score = 46%; and CANMAT [25], score = 54%. Table 1 briefly describes their characteristics.

### General recommendations according to the main topic for pharmacological treatment of depression

Only three CPGs provided information regarding the evaluation and management of the risk of suicide with pharmacological treatment, and all recommendations about this subject were considered strong [22,23,26]. Four CPGs recommended non-pharmacological treatment as

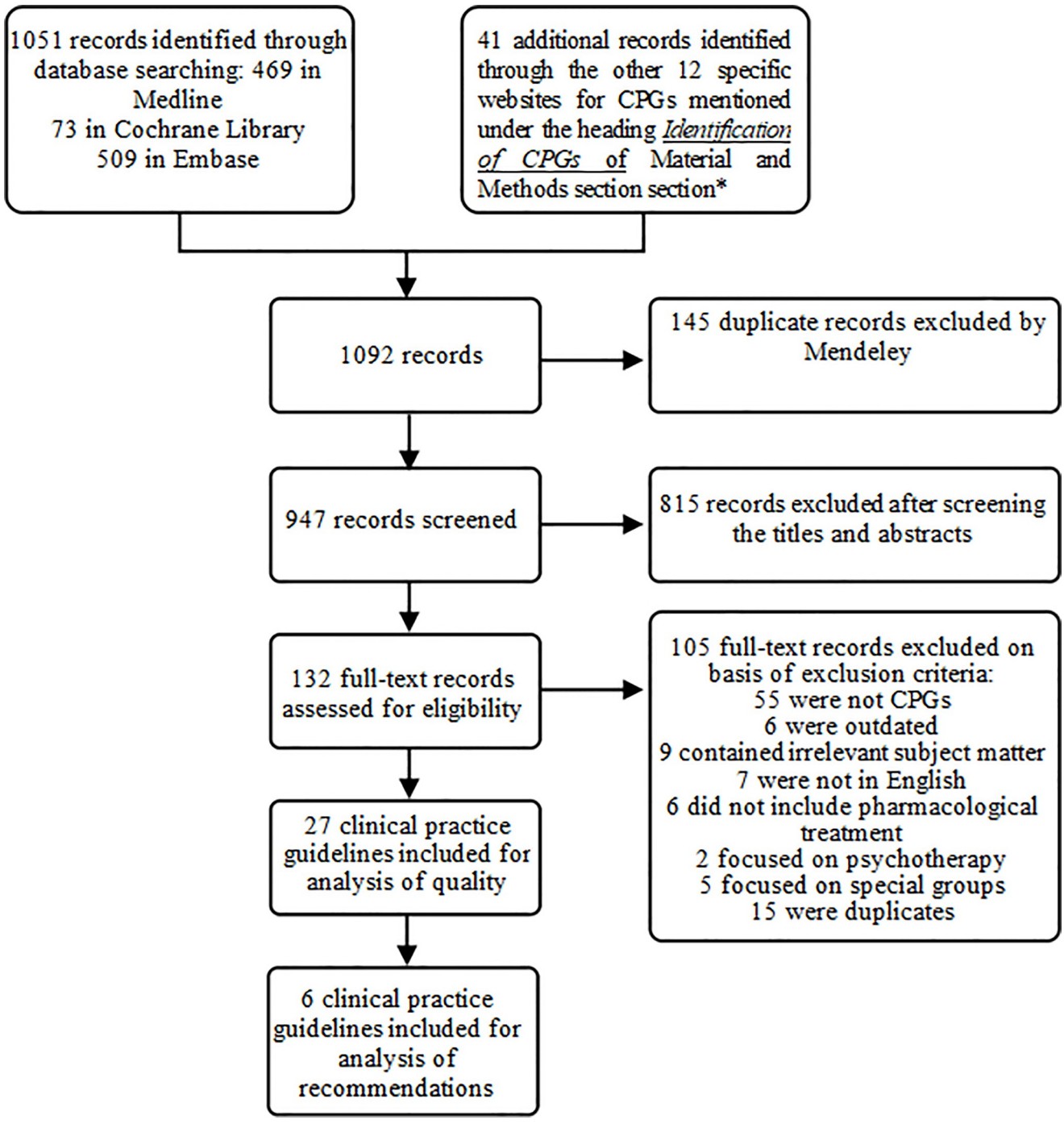

**Fig 1. Flowchart of clinical practice guidelines selection.**

first-line treatment for mild depression [21–23,25]. From the CPGs that mentioned recommendations about the indication of pharmacological treatment for patients with moderate or severe depression and a combination of pharmacological and psychotherapy treatments [21–23,26], such recommendations were considered strong (Table 2).

**Table 1. Clinical practice guidelines for the pharmacological treatment of depression included for analysis of recommendations.**

| Clinical practice guideline/Country/ Development organization | Main characteristics |
|---|---|
| Guía Clínica AUGE[3] [21] / Chile/ Ministry of Health) | This CPG[1] was prepared in 2013, after two previous versions developed in 2007 and 2009. It has a very well-organized format and flowcharts that allow a clear understanding of recommendations. There are 11 recommendations for pharmacological treatment. Among them, some are intended for specific populations that were not included in this study (e.g., pregnant women and adolescents). This CPG has a methodological manual that recommends the use of a GRADE[2] evidence classification system. However, it makes clear that a simplified system was used to classify the evidence, since the transition to the GRADE methodology is a gradual process that requires specific skills. |
| Guía de Práctica Clínica [22] / Colombia/ Ministry of Health | This CPG, prepared in 2013, used the adaptation of earlier versions of NICE[4] and CANMAT[5]. It is quite extensive, and its recommendations are organized into clinical questions. |
| Depression in adults [23] / England / NICE | This CPG was first published in 2009 and was updated in 2016. The full version of the document is quite extensive. It uses GRADE but does not contain the level of evidence or the strength of recommendation explicitly in the recommendations. In 2018, NICE updated this CPG exclusively to make recommendations regarding the use of valproate in pregnancy. These were not included in the current study. |
| Depression, adult in primary care [24] / USA / ICSI | This CPG comprises many documents. One of them contains the main recommendations clearly highlighted according to their level of evidence and strength of recommendation. Some recommendations are scattered in the text, and do not contain the level of evidence or strength of recommendation. |
| Practice guideline for the treatment of patients with major depressive disorder [26] / USA / APA | This is the third version of this CPG. APA reaffirmed this guideline in October 2015. It contains several detailed recommendations mainly related to subtypes of depression comorbidities and other clinical conditions associated with depression. It has a simplified evidence classification scale and a manual containing information about its elaboration and development. |
| Canadian Network for Mood and Anxiety Treatments 2016 [25] / CANADA / CANMAT | This CPG was created in 2001, followed by a subsequent update in 2009. It is currently in its third version. It comprises 21 questions followed by their respective answers. Researchers chose not to follow AGREE II for its development. The developers understood the wide international use of GRADE but opted instead for a self-classification evidence scale. This CPG also contains a separate document in which its development methods are described. |

[1]CPG = clinical practice guideline;

[2]GRADE = The Grading of Recommendations Assessment, Development and Evaluation;

[3]AUGE = Aseguramiento Universal de las Garantías Explícitas;

[4]NICE = National Institute for Health and Care Excellence;

[5]CANMAT = Canadian Network for Mood and Anxiety Treatments;

[6]ICSI = Institute for Clinical Systems Improvement;

[7]APA = American Psychiatry Association.

**Table 2. Clinical practice guidelines for the pharmacological treatment of depression: Recommendations.**

| Recommendations/clinical practice guidelines | Chilean CPG[1] [21] | Colombian CPG [22] | England CPG [23] | ICSI[2] CPG [24] | APA[3] CPG [26] | CANMAT[4] CPG [25] |
|---|---|---|---|---|---|---|
| **Risk of suicide** (related to pharmacotherapy) | - | ● | ● | - | ● | - |
| Frequent monitoring of patients who take antidepressants | - | ● | ● | - | - | - |
| Risk of overdose toxicity | - | ● | ● | - | - | - |
| Treatment suggestions for patients at high risk for suicide | - | - | - | - | ● | - |
| **Indication of pharmacological treatment** | ● | ● | ● | ● | ● | ● |
| Preference for non-pharmacological treatment for mild depression | ● | ● | ● | - | - | ● |
| Combination of antidepressants and psychotherapy whenever possible | - | - | - | ● | - | - |
| Combination of antidepressants and psychotherapy | | | | | | |
| *- in moderate depression* | - | ● | ● | | ● | - |
| *- in severe depression* | ● | ● | ● | - | ● | - |
| **First-line drug choice** | ● | ● | ● | ● | ● | ● |
| Selective serotonin reuptake inhibitors | ● | ● | ● | ● | ● | ● |
| Contraindication of tricyclics | ● | - | - | ● | - | - |
| Amitriptyline | - | ● | - | - | - | - |
| Mirtazapine | - | ● | - | ● | ● | ● |
| Agomelatine, mianserin, and milnacipran | - | - | - | - | - | ● |
| **Follow-up phase—duration of treatment after remission** | ● | ● | ● | - | - | ● |
| 6 months | ● | ● | ● | - | - | ● |
| 4–9 months | - | - | - | - | – | ● |
| **Maintenance phase—when deciding to prolong treatment (considerations)** | ● | ● | ● | - | ● | ● |
| Number of previous episodes | ● | ● | ● | - | ● | ● |
| Residual symptoms and coexisting conditions | - | ● | ● | - | ● | ● |
| Severity of symptoms | - | ● | ● | - | ● | - |
| Age | - | - | ● | - | ● | - |
| Frequency and persistence of symptoms | - | - | - | - | ● | - |
| Risk factors (lead to treatment for 2 years or longer) | - | ● | ● | - | - | ● |
| **Treatment discontinuation** | - | ● | ● | - | ● | - |
| Gradual suspension of antidepressants | - | ● | ● | - | ● | - |
| Need for a more progressive suspension for specific drugs (paroxetine and venlafaxine) | - | ● | ● | - | - | - |
| Informing patients about discontinuation symptoms | - | - | ● | - | ● | - |

[1]CPG = clinical practice guideline;

[2]ICSI = Institute for Clinical Systems Improvement;

[3]APA = American Psychiatry Association;

[4]CANMAT = Canadian Network for Mood and Anxiety Treatments.

● = CPG does contain topic; - = CPG does not contain topic.

All CPGs indicated serotonin selective reuptake inhibitors (SSRIs) as an option for first-line treatment for depression, and the recommendations were based on high-quality studies; however, most CPGs did not cite specific drugs. Mirtazapine was considered as first-line treatment by four CPGs [22,24–26]. The tricyclic amitriptyline, fluoxetine, sertraline, and mirtazapine were considered first-line drugs by the Colombian CPG—amitriptyline for patients without

**Table 3. Clinical practice guidelines for the pharmacological treatment of depression: Patients with a partial or no response.**

| Recommendations/clinical practice guidelines | Chilean CPG[1] [21] | Colombian CPG [22] | England CPG [23] | ICSI[2] CPG [24] | APA[3] CPG [26] | CANMAT[4] CPG [25] |
|---|---|---|---|---|---|---|
| **Strategies to improve the treatment for partial responders** | ● | ● | ● | ● | ● | ● |
| **Analysis of the factors contributing to an unsatisfactory response** | - | - | ● | - | ● | - |
| Check compliance to treatment and drugs dosage | - | - | ● | - | ● | - |
| Check for adverse effects caused by the drugs | - | - | ● | - | ● | - |
| **Adjusting the drugs dosage** | ● | ● | ● | ● | ● | ● |
| Consider age, coexisting conditions, concomitant pharmacological treatment, and adverse effects caused by the drugs | - | - | ● | - | ● | - |
| Consider increasing the dosage | ● | ● | ● | ● | ● | ● |
| **Replacing the drugs** | ● | ● | ● | ● | ● | ● |
| Consider replacing the drugs | ● | ● | ● | ● | ● | ● |
| Replacement: | | | | | | |
| *- at first, within the same antidepressant class* | - | ● | ● | - | - | - |
| *- a different class of antidepressants (* | - | ● | - | - | - | ● |
| Amitriptyline as a second-line strategy | - | ● | ● | - | - | ● |
| **Combination or augmentation drugs** | ● | ● | ● | ● | ● | ● |
| Antipsychotic agents | - | ● | ● | ● | ● | ● |
| Monoamine oxidase inhibitors | - | ● | - | - | ● | - |
| Olanzapine | - | - | ● | ● | - | ● |
| Mirtazapine | - | - | ● | ● | - | ● |
| Risperidone | - | ● | ● | - | - | ● |
| Lithium | ● | ● | ● | - | ● | ● |
| Thyroid hormones | ● | - | - | - | ● | - |

[1]CPG = clinical practice guideline;

[2]ICSI = Institute for Clinical Systems Improvement;

[3]APA = American Psychiatry Association;

[4]CANMAT = Canadian Network for Mood and Anxiety Treatments.

● = CPG does contain topic; - = CPG does not contain topic.

contraindications and the others for patients with contraindications to tricyclics. The recommendations were based on pharmacoeconomic studies. Sertraline and mirtazapine were considered owing to cost per quality-adjusted life years [22]. The CANMAT CPG recommended agomelatine, mianserin, and milnacipran as first-line treatment [25]. More details are available in S1 Table.

## Recommendations for treatment of depression in those who did not respond or partially responded

All CPGs included recommendations for the treatment of those who did not respond or partially responded to first-line therapy. Such recommendations are synthesized in Table 3 and details are presented in S2 Table. Almost all recommendations were considered strong regarding the adjustment of drug dosages when there was a lack of response to the initial pharmacological treatment. Moreover, antipsychotic agents were recommended as an augmentation strategy by five CPGs.

However, considering the differences found among the CPGs, the CANMAT was the only one that grouped the recommendations for adjuvant treatment (i.e., combination and augmentation strategies), in first-, second-, and third-line strategy. Moreover, the level of evidence

and strength of recommendations for the combination and augmentation strategies varied among the CPGs. Although the same evidence scale was not precisely used to classify among the five CPGs, the evidence for recommendation of antipsychotic agents for augmentation was considered with substantial clinical safety by the APA CPG, weakly by the Colombian CPG. The remaining CPGs did not specify the level of the evidence or the strength of the recommendations.

Among the five CPGs that included lithium as an augmentation strategy, only three clearly indicated the strength of recommendation; however, discrepancies existed among them. While the Chilean CPG strongly recommended it, the Colombian CPG recommendation was weak, and the APA CPG recommended it, based on moderate clinical safety. Considering thyroid hormones as an augmentation strategy, two CPGs included it; however, they disagreed on the strength of the recommendation [21,26]. The Chilean CPG strongly recommended thyroid hormones, while the APA CPG indicated that there was moderate evidence of clinical safety.

### Recommendations for treatment of depression considering its subtypes

Except for the Chilean CPG, all included recommendations per depression subtypes (Table 4). Moreover, most CPGs [22,23,25,26] provided recommendations for treating psychotic depression, except the ICSI and the Chilean CPG. The recommendations for treating depression with psychotic features was the use of antipsychotic and antidepressants agents. Regardless, the classification method applied to the significance of the recommendation or the quality of the evidence, using these drugs was considered a strong recommendation or was based on high-quality evidence.

No high-quality CPG mentioned the treatment of catatonic depression. On the other hand, the most employed guidelines in clinical practice—CANMAT and APA CPG—recommended using benzodiazepines for the treatment of catatonic depression; however, they disagreed on the classification of the quality of the evidence [25,26]. Moreover, depression with atypical characteristics was considered in the CPGs most used in clinical practice and in the ICSI CPG [24–26]. Depression with melancholic characteristics was contemplated only in the CPGs most used in clinical practice [25,26]. Seasonal, with somatic symptoms, and mixed depression disorders were only addressed in the CANMAT CPG [25]. More details are shown in S3 Table.

## Discussion

First, it should be noted that pharmacological treatment of depression is one of the strategies that should be considered to ensure adequate patient care. Pharmacotherapy should be prescribed only after a careful evaluation of the patient, including risk of suicide, requirement of hospitalization, indication of psychotherapy, and existence of comorbidities among other clinical and psychosocial aspects.

Relevant findings of this study include the absence of specific recommendations regarding suicide risk associated with pharmacotherapy in some CPGs [21,24,25]; convergence on SSRIs as first-line antidepressants by all CPGs, aside with indication of some peculiar antidepressants as first-line by two CPGs [22,25]; specific strategies for treatment of a partial response are missing in various CPGs; and treatment specificities for depression subtypes being covered by all CPGs except one [21].

Recommendations for pharmacological treatment of depression.

### Suicidalit

Depression is considered the principal risk factor for suicide, and suicide risk peaks in the first weeks of antidepressant treatment [27]. However, only three of the 6 CPGs included a topic

**Table 4. Clinical practice guidelines for the pharmacological treatment of depression: Subtypes of depression.**

| Recommendations/clinical practice guidelines | Chilean CPG[1] [21] | Colombian CPG [22] | England CPG [23] | ICSI[2] CPG [24] | APA[3] CPG [26] | CANMAT[4] CPG [25] |
|---|---|---|---|---|---|---|
| **Subtypes of depression** | - | ● | ● | ● | ● | ● |
| **Chronic or dysthymia** | - | - | - | ● | - | - |
| Combination of antidepressants and psychotherapy | - | - | - | ● | - | - |
| Beginning of pharmacological treatment (pure dysthymia) | - | - | - | ● | - | - |
| **Depression with psychotic features** | - | ● | ● | - | ● | ● |
| Contraindication of isolated psychotherapy | - | - | - | - | - | - |
| Indication of antipsychotic agents in combination with antidepressants | - | ● | ● | - | ● | ● |
| **Catatonic** | - | - | - | - | ● | ● |
| Benzodiazepines in combination with antidepressants | - | - | - | - | ● | ● |
| Barbiturates in combination with antidepressants | - | - | - | - | ● | - |
| **Depression with atypical features** | - | - | - | ● | ● | ● |
| Monoamine oxidase inhibitors | - | - | - | ● | ● | - |
| Selective serotonin reuptake inhibitors and bupropion | - | - | - | - | ● | - |
| No specific antidepressants have demonstrated superiority | - | - | - | - | - | ● |
| **Depression with melancholic features** | - | - | - | - | ● | ● |
| Serotonin and noradrenaline reuptake inhibitors and tricyclic antidepressants | - | - | - | - | ● | - |
| No specific antidepressants have demonstrated superiority | - | - | - | - | - | ● |
| **Depression with seasonal pattern** | - | - | - | - | - | ● |
| No specific antidepressants have demonstrated superiority | - | - | - | - | - | ● |
| **Depression with somatic symptoms (fatigue)** | - | - | - | - | - | ● |
| **Mixed depression disorder** | - | - | - | - | - | ● |

[1]CPG = clinical practice guideline;

[2]ICSI = Institute for Clinical Systems Improvement;

[3]APA = American Psychiatry Association;

[4]CANMAT = Canadian Network for Mood and Anxiety Treatments.

● = CPG does contain topic; - = CPG does not contain topic.

concerning the risk of suicide associated with pharmacotherapy [22,23,26]. Thus, our findings highlight that future CPGs should consider the risk of suicide associated with pharmacological treatment.

## First-line treatment

All CPGs considered SSRIs as a first-line antidepressant treatment. However, we identified two important discrepancies. Besides SSRIS, as options, the CANMAT CPG recommended the use of agomelatine, milnacipran, and mianserin [25]; and the Colombian CPG recommended the use of amitriptyline [22] as first-line treatment based on pharmacoeconomic studies.

Discrepancies among recommendations can negatively affect healthcare professionals' confidence in CPGs [28]. We found such differences even among high-quality CPGs. CPGs may provide different recommendations according to cultural differences or based on the availability and infrastructure of a healthcare system [29]. Consequently, values shared by developers and patients, aside from cost issues, may influence the choice of a recommended treatment

and reduce the reliability of the recommendation, particularly when scientific evidence is weak [30–32].

According to the CANMAT CPG [25], agomelatine demonstrated favorable efficacy and tolerability in a network meta-analysis of new-generation antidepressants conducted by Khoo et al (2015) [33]. The advantage of network meta-analysis is that multiple treatments can be subjected to both direct (i.e., among randomized controlled trials) and indirect (i.e., across trials based on a common variable) comparisons of interventions; the effects of different interventions that have not been investigated in trials can be compared, and the analysis of all interventions enables ranking of therapeutic alternatives with regard to a given outcome. However, other aspects must be taken into account: To assess the level of evidence or strength of the recommendation, it is important to consider not only the type of study but also its quality or risk of bias, or both, as recommended in the GRADE method, for example [29,34]. Network meta-analysis should be conducted under strict and specific methodology conditions (transitivity and consistency criteria met); the inclusion of studies at risk of bias may negatively compromise the validity of the findings [35]. In their network meta-analysis, Khoo et al did not describe the result of the risk of bias assessment for any of the eight clinical trials included in the meta-analysis, but in the general assessment, most of the primary studies included were judged as having high or unclear risk of bias in at least one domain of the Cochrane Collaboration risk of bias tool [33]. In a 2013 Cochrane review of 13 trials in a pairwise meta-analysis [36] and in a 2018 network meta-analysis conducted by Cipriani et al [3], agomelatine did not show robust advantages over the other antidepressants for effectiveness, but it appeared to be better tolerated. In both studies, the limitation was that the primary studies had biases ([3]). In addition, agomelatine was not approved by the U.S. Food and Drug Administration to treat depression, and its safety and efficacy have been questioned [36]. The Colombian CPG explicitly recommends not using agomelatine because of insufficient evidence on its effectiveness [22].

The CANMAT CPG recommendation of milnacipran as a first-line treatment is based more on its tolerability than on its efficacy [37], which does not follow the recommended sequence for the rational use of medicines: efficacy, availability, and safety [38]. Moreover, in Cipriani et al's study [3], milnacipran did not stand out in terms of either effectiveness or safety [3]. Milnacipran is also not approved by the U.S. Food and Drug Administration to treat depression [39].

The recommendation of mianserin as a first-line option is controversial because the CANMAT CPG itself cites the results of a network meta-analysis that demonstrated few differences in response, although SSRIs and TCAs were superior to mianserin/mirtazapine and moclobemide [25,40]. Indeed, Linde et al [40] affirmed that physicians should be aware that SSRIs and TCAs have a somewhat more solid evidence base than do other pharmacological classes. Cipriani et al [3] did not include mianserin in network meta-analysis [3]. Arroll et al [41] conducted a meta-analysis of two trials in which mianserin was compared with placebo and concluded that mianserin was effective for continuous outcomes but did not affect rates of remission and response [41].

The Colombian CPG inclusion of amitriptyline (for patients without a contraindication to CTAs) as a first-line option was based on its good cost-effectiveness profile. In the pharmacoeconomic study conducted by the Colombian CPG, sertraline was considered more cost-effective than amitriptyline regarding outcome quality–adjusted life years. With regard to its efficacy, a meta-analysis supported the superiority of amitriptyline over other tricyclic/heterocyclic antidepressants and over SSRIs; however, the effect size was clinically not relevant [42]. Cipriani et al [3] confirmed the effectiveness of amitriptyline but revealed that it had one of the highest rates of treatment abandonment owing to adverse events [3]. In addition, evidence

regarding the tolerability and risk of CTAs is plentiful and includes a narrow safety margin and considerable risk of death in cases of overdose [43]. With regard to individuals at risk for adverse events, such as older adults (i.e., the Beers criteria), the ICSI CPG stresses that tricyclics must not be used by older adults because of their anticholinergic effects and their capacity to induce orthostatic hypotension and stimulation of the central nervous system [44].

Despite recommending SSRIs as first-line treatments, most CPGs did not cite specific SSRIs. Only three CPGs specifically indicated the SSRIs used as the first-line treatment [22,24,25]. Although only a few controlled randomized trials have compared SSRIs head-to-head [3], it is difficult to explain why some CPGs recommend specific SSRIs while others recommend a group of SSRIs.

## Partial and non-responders

The initial pharmacological treatment for depression had a response rate ranging from 40–60% [45], and only around 30% achieved remission [45,46]. Consequently, recommendations to non-responders should be an essential part of any CPG. In fact, all the present CPG recommendations addressed first-line therapy and non-responders. Adjustment of the dosage with strong evidence was a consensus among the CPGs.

It is worth noting that many CPGs describe the alternatives of replacing the antidepressant for those not responding to first line treatment without specifying which should be considered the second, the third or the fourth-line therapy. Additionally, among those specifying a sequence of strategies, the concept and support to define each line of therapy varies across the CPGs, leading to considerable discrepancies. For example, the Colombian CPG, instead of including other antidepressants, recommends using as second-line treatment, first-line alternatives such as fluoxetine, sertraline, amitriptyline or mirtazapine, that had not been prescribed. Regarding third-line treatment, the Colombian CPG recommends imipramine, clomipramine, paroxetine, escitalopram, citalopram, fluvoxamine, venlafaxine, duloxetine, desvenlafaxine, trazodone, and bupropion [22], while third line alternatives for the Canadian CPG (CANMAT) are Monoamine oxidase inhibitors (i.e. Phenelzine and tranylcypromine) and reboxetine [25]. On the other hand, for the APA CPG, MAOIs are considered fourth line therapy [26].

Also recommended by most CPGs was the augmentation with antipsychotics for partial responders. However, some CPGs did not report, while others disagreed, about the strength of recommendation or level of evidence for augmentation strategies. Furthermore, the CANMAT CPG was the only one recommending an order for the augmentation strategies.

## Subtypes of depression

Depression with psychotic features was the subtype receiving most consistent recommendations across the CPGs. Four CPGs recommended antipsychotic agents in combination with antidepressants for the treatment of depression with psychotic features [22,23,25,26]. Various studies, including meta-analyses, have supported the advantage of associating antidepressants with antipsychotics for the treatment of depression with psychotic features [47]. The goal of the ICSI CPG was primary-care attention, which explains its absence of recommendations for depression with psychotic features. The Chilean CPG included only comments about the effectiveness of antidepressants and antipsychotics for psychotic depression; however, it is outside the topic of recommendations [21].

Data supporting distinct antidepressant response of atypical depression to monoamine oxidase inhibitors (MAOIs; i.e., tranylcypromine) have been reported since the 1960s [48]. The presence of melancholic or atypical features was considered respectively by two and three

CPGs. The APA CPG was the only CPG that addressed specific treatment for melancholic depression—serotonin and noradrenaline reuptake inhibitors (SSRIs) and tricyclic antidepressants; the CANMAT CPG stated that no antidepressant has been proven to be superior for melancholic depression. For depression with atypical features, the APA CPG and the ICSI CPG recommended MAOIs, while the CANMAT CPG stated that no antidepressant has been proven to be superior. Such discrepancies are in line with controversies in the literature. Notwithstanding the traditional view of superiority of MAOIs for depression with atypical features, the iSPOT-D Trial compared the efficacy of venlafaxine, escitalopram, and sertraline and found similar symptom reduction trajectories among depression subtypes including melancholic, atypical, anxious depression, and subtype combinations [49].

Considering catatonic depression, benzodiazepines may lead to a rapid relive for some patients [50]; however, only the APA and the CANMAT CPGs included such recommendation. Although, it can be argued that catatonic depression is not frequent, it has a high morbidity and our view is that consideration to specificities of its treatment should be included in CPGs.

## Strengths and limitations

We included CPGs published in English, Portuguese, and Spanish; thus, the results might not reflect relevant CPGs in other languages. The intrinsic subjectivity limitation of the AGREE II should also be considered; to minimize this, we included three evaluators and disagreements among them were solved by discussion until consensus was reached. We also included two additional CPGs based on their wild acceptability (CANMAT and APA CPG) [25,26] instead of basing the study only on the AGREE II evaluation. Another point to consider is that some recommendations might have not been considered because they were placed outside the topic of pharmacological treatment or it was not clear if they were recommendations or mere dissecting of evidence.

Among the strengths of this study, we cite the comprehensive search and the careful training of appraisers. In describing the recommendations, we offer a comparative view of distinct CPGs that provide physicians and patients a more comprehensive understanding of pharmacological approaches to the treatment of depression. Our findings could help in the elaboration/adaptation of a CPG because we identified important divergences among existing CPGs to which stakeholders (patients and professionals) should give special attention. Moreover, the identification of the points at which CPGs converge fully and that have been well addressed in certain CPGs may also be helpful for the elaboration/adaptation of a CPG for local contexts, as well as contribute to clinical decisions about treatment for this severe mental health problem.

## Conclusion

We found that the various CPGs were typically consistent with each other; however, they presented some vital differences. The use of SSRIs as first-line pharmacological treatment and its dose adjustment for non-responsive patients were consistent among all the included CPGs. Largely consistent was the use of antipsychotics as augmentation (except for the ICSI CPG, which addressed primary care) for non-responders. Importantly, only 50% of the CPGs addressed the risk of suicide associated with pharmacotherapy. Considering the increased risk of suicide associated with the first few weeks of antidepressant treatment, recommendations regarding this topic should be mandatory in all CPGs. Moreover, specificities for some subtypes of depression (e.g., catatonic and atypical) were addressed by some but not all CPGs. Differences in the level of evidence or strength of recommendation were very frequent among the CPGs, and some of them presented unique recommendations. These findings support that,

when using a specific guideline for the treatment of depression, caution is needed to provide the most appropriate treatment to each patient.

## Supporting information

**S1 Appendix. Systematic review strategy: Research terms used to identify clinical practice guidelines in the medline, embase, and the Cochrane Library databases.**
(DOCX)

**S2 Appendix. Reasons for the inclusion and exclusion of clinical practice guidelines.**
(DOCX)

**S3 Appendix. Quality assessment of the included clinical practice guidelines.**
(DOCX)

**S1 Table. Recommendations for the treatment of depression, extracted from clinical practice guidelines, listed for elaborating the synthesis published between January 2011 and April 2019: Indications and strategies of pharmacological treatment.**
(DOCX)

**S2 Table. Recommendations for the treatment of depression, extracted from clinical practice guidelines, listed for elaborating the synthesis published between January 2011 and April 2019: Patients who did not respond or partially responded.**
(DOCX)

**S3 Table. Recommendations for the treatment of depression, extracted from clinical practice guidelines, listed for the elaboration of the synthesis published between January 2011 and April 2019: Treatment for subtypes.**
(DOCX)

**S1 Checklist. PRISMA 2009 checklist.**
(DOC)

## Acknowledgments

We would like to acknowledge everyone who contributed their time and knowledge to this study without any financial support. We would like to thank CHRONIDE for their support in the appraisal of CPGs. We also appreciate the assistance of Professor Carlota de Oliveira Rangel Yagui in manuscript preparation. We are indebted to Caroline Molino, Luciana Vasconcelos, and Sheila Kalb Wainberg for their invaluable assistance in helping us understand the method of appraisal of CPGs. We also recognize Andrea Fernandes Larruscain and Aliandra Fantinell de Oliveira for their help with manuscript formatting. Finally, we are grateful to Alfredo Jose Neto for his assistance in discussion of data synthesis and Carlos Eduardo Moscato Fuzaro for his support in data.

## Author Contributions

**Conceptualization:** Franciele Cordeiro Gabriel, Daniela Oliveira de Melo, Renério Fráguas, Nathália Celini Leite-Santos, Rafael Augusto Mantovani da Silva, Eliane Ribeiro.

**Data curation:** Franciele Cordeiro Gabriel, Daniela Oliveira de Melo, Renério Fráguas, Nathália Celini Leite-Santos, Rafael Augusto Mantovani da Silva, Eliane Ribeiro.

**Formal analysis:** Franciele Cordeiro Gabriel, Daniela Oliveira de Melo, Renério Fráguas, Eliane Ribeiro.

**Funding acquisition:** Franciele Cordeiro Gabriel, Eliane Ribeiro.

**Investigation:** Franciele Cordeiro Gabriel, Daniela Oliveira de Melo, Renério Fráguas, Eliane Ribeiro.

**Methodology:** Franciele Cordeiro Gabriel, Daniela Oliveira de Melo, Renério Fráguas, Nathália Celini Leite-Santos, Eliane Ribeiro.

**Project administration:** Franciele Cordeiro Gabriel, Daniela Oliveira de Melo, Eliane Ribeiro.

**Resources:** Franciele Cordeiro Gabriel, Daniela Oliveira de Melo, Eliane Ribeiro.

**Software:** Franciele Cordeiro Gabriel, Daniela Oliveira de Melo, Eliane Ribeiro.

**Supervision:** Franciele Cordeiro Gabriel, Daniela Oliveira de Melo, Eliane Ribeiro.

**Validation:** Franciele Cordeiro Gabriel, Daniela Oliveira de Melo, Renério Fráguas, Eliane Ribeiro.

**Visualization:** Franciele Cordeiro Gabriel, Daniela Oliveira de Melo, Renério Fráguas, Eliane Ribeiro.

**Writing – original draft:** Franciele Cordeiro Gabriel, Daniela Oliveira de Melo, Renério Fráguas, Eliane Ribeiro.

**Writing – review & editing:** Franciele Cordeiro Gabriel, Daniela Oliveira de Melo, Renério Fráguas, Nathália Celini Leite-Santos, Rafael Augusto Mantovani da Silva, Eliane Ribeiro.

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
