## [Decision Letter · Decision Letter 0]

7 Jan 2020

PONE-D-19-16309

Pharmacological treatment of depression: a systematic review comparing clinical practice guideline recommendations

PLOS ONE

Dear MSc Gabriel,

Thank you for submitting your manuscript to PLOS ONE. After careful consideration, we feel that it has merit but does not fully meet PLOS ONE’s publication criteria as it currently stands. Therefore, we invite you to submit a revised version of the manuscript that addresses the points raised during the review process.

We would appreciate receiving your revised manuscript by Feb 21 2020 11:59PM. To enhance the reproducibility of your results, we recommend that if applicable you deposit your laboratory protocols in protocols.io, where a protocol can be assigned its own identifier (DOI) such that it can be cited independently in the future. For instructions see: http://journals.plos.org/plosone/s/submission-guidelines#loc-laboratory-protocols

We look forward to receiving your revised manuscript.

Kind regards,

Gabriele Fischer, MD

Academic Editor

PLOS ONE

Additional Editor Comments:

The authors present a detailed picture of the current status of CPGs for the treatment of depression. Please pay particular attention in addressing the reviewers comments point by point. The raised concern of reviewer 2 that the study should be described as a descriptive study of an existing data set, rather than a systematic review should be emphasized. Further, implications for guideline developers which arise from the authors work should be discussed in more detail.

Journal Requirements:

'FCG NCLS

This study was financed in part by the Coordenação de Aperfeiçoamento de Pessoal de Nível Superior - Brasil (CAPES) - Finance Code 001"

https://www.capes.gov.br/pt/

The funders had no role in study design, data collection and analysis, decision to publish, or preparation of the manuscript.'

a. Please provide an amended statement that declares *all* the funding or sources of support (whether external or internal to your organization) received during this study, as detailed online in our guide for authors at http://journals.plos.org/plosone/s/submit-now.  

Please also include the statement “There was no additional external funding received for this study.” in your updated Funding Statement.

3. We note you have included a table to which you do not refer in the text of your manuscript. Please ensure that you refer to Table 4 in your text; if accepted, production will need this reference to link the reader to the Table.

Reviewers' comments:

Reviewer's Responses to Questions

**Comments to the Author**

1. Is the manuscript technically sound, and do the data support the conclusions?

Reviewer #1: Yes

Reviewer #2: Partly

2. Has the statistical analysis been performed appropriately and rigorously? 

Reviewer #1: Yes

Reviewer #2: N/A

3. Have the authors made all data underlying the findings in their manuscript fully available?

Reviewer #1: Yes

Reviewer #2: Yes

4. Is the manuscript presented in an intelligible fashion and written in standard English?

Reviewer #1: Yes

Reviewer #2: Yes

5. Review Comments to the Author

Reviewer #1: Major depressive disorder is one of the most prevalent and life-threatening forms of mental illnesses and a major cause of morbidity worldwide. Clinical practice guidelines for depression represent an essential tool to improve patient healthcare by outlining practices recommended based on scientific research.

The manuscript entitled “ Pharmacological treatment of depression: a systematic review comparing clinical practice guideline recommendations ” examines the most relevant CPGs for the pharmacological treatment of depression and identifies a matrix of recommendations including agreements and potential disagreements among CPGs.

The authors found that all CPGs indicated serotonin selective reuptake inhibitors (SSRIs) as an option for first-line treatment for depression, whereas only three CPGs provided information regarding the evaluation and management of the risk of suicide with pharmacological treatment.

In general, the conclusions are supported by the experimental data and this manuscript provides a more comprehensive view of pharmacological approaches to the treatment of depression.

Major

Currently available antidepressants are effective for most patients, but around 30% are considered treatment resistant (TRD), a condition that is associated with a significant impairment of cognitive function and poor quality of life. The authors found that the level of evidence and strength of recommendations for the combination and augmentation strategies varied among the CPGs. Have the authors found any recommendation in analyzed CPGs for the treatment of depressed patients who did not respond to second-line therapy (TRD patients) ?

Reviewer #2: Thank you for the opportunity to review this manuscript. It is well written and provides a clear description the current status of high-quality CPGs for the treatment of depression and provides useful insight into the differences between key guidelines.

Methods:

This manuscript is based on an data from an existing systematic review (SR) and the authors mention this study as a secondary analysis. I would thus argue that this study is not a systematic review, but rather a descriptive study of an existing dataset (from the previous SR). S1-2 is also thus not needed as these are simply outputs from the previously published SR.

In order for this study to qualify as a SR, some form of synthesis should occur, the process of which should be well-described.

Thus the method of reporting for this manuscript would be better suited using STROBE than PRISMA.

Figure 1 is labelled as a synthesis of CPGs. However, how this synthesis was done is unclear and should be elaborated. At face value, Figure 1 is simply a decision algorithm that provides an vague overview of clinician considerations in patients with suspected depression.

Presently, the authors in Tables 1-4 describe the landscape of recommendations across different CPGs grouped by categories, and further elaborate on this in the results section. This is critical realising the authors aim establishing the extent and preventing redundant CPG efforts.

Discussion

The authors identified useful gaps in CPG recommendations and provide a useful comparative discussion for future CPG developments.

Recommendations clearly differed across CPGs, as mentioned, however some CPGs used evidence from standard SR and pair-wise meta-analysis (MA) while others used network MA, which is able to compare multiple treatments. This more appropriate method of evidence synthesis and it's impact on CPG recommendations and as a contributing factor or explanation for the heterogeneity in results should be discussed. Especially as NMA is only recently becoming mainstay in evidence for CPGs

Linked to the comments above, the authors, as part of their aim highlight the importance of their work for LMICs guideline developers. However, this is not demonstrated or elaborated on further in the manuscript. Do consider placing the research findings for LMICs as stated in the objectives.

The authors mention guideline adaptation, it would be useful to include what these results mean for guideline developers who would use adaptation methods (e.g. those with limited resources, not just in LMICs), what challenges it poses and what further specific research would be needed to create trustworthy CPG considering your results.

6. PLOS authors have the option to publish the peer review history of their article (what does this mean?). If published, this will include your full peer review and any attached files.

Reviewer #1: No

Reviewer #2: No

---

## [Author Response · Author response to Decision Letter 0]

18 Feb 2020

1) Editor's Comments:

“The authors present a detailed picture of the current status of CPGs for the treatment of depression.”

Author’s reply: Thank you!

“Please pay particular attention in addressing the reviewer’s comments point by point. The raised concern of reviewer 2 that the study should be described as a descriptive study of an existing data set, rather than a systematic review should be emphasized.”

Author’s reply: We carefully revised the manuscript thoroughly and e noted all the corresponding modifications. Moreover, considering the concern raised by reviewer 2, we believe that our study fulfills the criteria for a systematic review. However, we agree that we did not describe the methodology clearly in the previous version of the manuscript. In the revised version, we described the procedures that we performed in the systematic review process.

“Further, implications for guideline developers which arise from the authors work should be discussed in more detail.”

Author’s reply: Thank you for this suggestion. We have added the following paragraph in the Discussion section: “Our findings could help in the elaboration/adaptation of a CPG because we identified important divergences among existing CPGs to which stakeholders (patients and professionals) should give special attention. Moreover, the identification of the points at which CPGs converge fully and that have been well addressed in certain CPGs may also be helpful for the elaboration/adaptation of a CPG for local contexts, as well as contribute to clinical decisions about treatment for this severe mental health problem” [page 26, lines 378-384].

2) Journal Requirements

“Please ensure that your manuscript meets PLOS ONE's style requirements, including those for file naming. The PLOS ONE style templates can be found at …”

Author’s reply: Thank you; we have done this.

“Thank you for stating in your Funding Statement: … Please also include the statement “There was no additional external funding received for this study.” in your updated Funding Statement…. Please include your amended Funding Statement within your cover letter. We will change the online submission form on your behalf.

Author’s reply: Thank you for your assistance. We have changed the funding statement from “This study was financed in part by the Coordenação de Aperfeiçoamento de Pessoal de Nível Superior - Brasil (CAPES) - Finance Code 001” to “This study was financed by the Coordenação de Aperfeiçoamento de Pessoal de Nível Superior - Brasil (CAPES) - Finance Code 001. No additional external funding was provided for this study.” Moreover, we added this funding statement in the cover letter.

“We note you have included a table to which you do not refer in the text of your manuscript. Please ensure that you refer to Table 4 in your text; if accepted, production will need this reference to link the reader to the Table.”

Authors’ reply: We are grateful for this observation. We have added the citation of Table 4 on page 17, line 204.

3) Reviewer #1 

Author’s comment: Firstly, thank you for sharing all your thoughts about our manuscript. We are glad that you think our findings are relevant. Please find below your comments and our answers: 

“Have the authors found any recommendation in analyzed CPGs for the treatment of depressed patients who did not respond to second-line therapy (TRD patients)?”

Authors’ reply: The recommendations for patients who are resistant to pharmacotherapy, including those who have not responded to second-line treatment, are contained in the supplemental Table S2 (patients who did not respond or showed partial response). To improve this point in the manuscript, we have included the following paragraph in the Discussion section” [page 23-24, lines 318-330] “It is worth noting that many CPGs describe the alternatives of replacing the antidepressant for those not responding to first line treatment without specifying which should be considered the second, the third or the fourth-line therapy. Additionally, among those specifying a sequence of strategies, the concept and support to define each line of therapy varies across the CPGs, leading to considerable discrepancies. For example, the Colombian CPG, instead of including other antidepressants, recommends using as second-line treatment, first-line alternatives such as fluoxetine, sertraline, amitriptyline or mirtazapine, that had not been prescribed. Regarding third-line treatment, the Colombian CPG recommends imipramine, clomipramine, paroxetine, escitalopram, citalopram, fluvoxamine, venlafaxine, duloxetine, desvenlafaxine, trazodone, and bupropion [22], while third line alternatives for the Canadian CPG (CANMAT) are Monoamine oxidase inhibitors (i.e. Phenelzine and tranylcypromine) and reboxetine [25]. On the other hand, for the APA CPG, MAOIs are considered fourth line therapy [26]”

Reviewer #2

“Thank you for the opportunity to review this manuscript. It is well written and provides a clear description the current status of high-quality CPGs for the treatment of depression and provides useful insight into the differences between key guidelines.”

Authors’ reply: Thank you for your comments. We are glad that you found our manuscript well written and clear. Please find your comments and our detailed replies listed below:

“This manuscript is based on data from an existing systematic review (SR) and the authors consider this study as a secondary analysis. I would thus argue that this study is not an SR but rather a descriptive study of an existing dataset (from the previous SR). S1-2 are also not needed as these are simply outputs from the previously published SR. For this study to qualify as an SR, some form of synthesis should occur, the process of which should be well-described. Thus, the suitable method of reporting for this manuscript involved STROBE than PRISMA.”

Authors’ reply: We understand what reviewer 2 has pointed out. We believe that the way we previously described the methods in this manuscript and the relationship between this study and the previous study was unclear which led to this question. In the previous study, we analyzed quality aspects of CPGs that were about 15 chronic diseases, and performed a specific systematic review procedure for each of the disease. Depression was only one of those chronic diseases. Furthermore, in the previous study, factors associated with the quality of CPGs for all 15 chronic diseases were appraised without the analysis of the recommendations. In this study, we analyzed the recommendations of the systematic review of the CPGs for depression. We changed some parts of the Methods section in order to make the review of the systematic procedures clearer, and detailed. Moreover, we also modified some points in the Result section: Table S2 in the revised manuscript is refers to the list of the guidelines, excluded through the reading of the whole text, and the flowchart that previously was S2 is now Fig 1. In addition, we also changed one sentence in the Results section in this regard.

Modifications done in Methods section: [page 4, lines 79-93]

“We recently reported CPGs for the pharmacological treatment of noncommunicable diseases that could be considered “high-quality” [12]. In that study, we conducted individual systematic reviews for each included disease; and using the second version of the Appraisal of Guidelines for Research and Evaluation (AGREE II), evaluated 421 CPGs to establish the quality of their protocol registered on PROSPERO (CRD42016043364) [12]. In this study, we focus specifically on the part of that systematic review about the pharmacological treatment of depression [12].

We conducted a comprehensive search in MEDLINE, Embase, and the Cochrane Library, as well as in 12 specific websites for CPGs, because all such databases are well-recognized guideline repositories that have been cited frequently in previous studies of systematic reviews [12, 13]. The CPG searched were published between 2011 and 2016 (details of search strategies are in S1 Appendix). In April 2019, we searched the literature to update the included CPGs. Two independent reviewers screened the records regarding the eligibility criteria and conducted the data extraction. Discrepancies were solved by consensus”.

Modifications done in Results section: [page 6, lines 132-135] “In our initial search, we identified 947 citations and abstracts after removing duplicates. Thereafter, by reading the full text and applying the eligibility criteria, we selected 27 CPGs for this study (Fig 1). (S2 Appendix includes the reason for excluding 105 full records)”.

“Figure 1 is labelled as a synthesis of CPGs. However, how this synthesis was done is unclear and should be elaborated. At face value, Figure 1 is simply a decision algorithm that provides a vague overview of clinician considerations in patients with suspected depression.”

Authors’ reply: Thank you for the opportunity to elaborate on this point. We based the design of Figure 1 on the CPGs analyzed together with discussions with a mental health provider who has specific expertise related to depression. The information in the original Figure 1 was a synthesis of an initial approach to the treatment of depression and provided critical information on the consideration of pharmacotherapy. However, we do agree that this information was primarily general in nature and that it did not add anything truly comprehensive to the understanding of our findings. As such, we have removed it from the revised manuscript.

“Presently, the authors in Tables 1-4 describe the landscape of recommendations across different CPGs grouped by categories, and further elaborate on this in the results section. This is critical realising the authors aim establishing the extent and preventing redundant CPG efforts. The authors identified useful gaps in CPG recommendations and provide a useful comparative discussion for future CPG developments.”

Authors’ reply: Yes, thank you very much for mentioning this. We hope that our revised manuscript is a reliable source for providing information and a useful tool to help reduce redundant efforts.

“Recommendations clearly differed across CPGs, as mentioned, however some CPGs used evidence from standard SR and pair-wise meta-analysis (MA) while others used network MA, which is able to compare multiple treatments. This more appropriate method of evidence synthesis and it's impact on CPG recommendations and as a contributing factor or explanation for the heterogeneity in results should be discussed. Especially as NMA is only recently becoming mainstay in evidence for CPGs.”

Authors’ reply: Thank you for this suggestion. We have included substantial argumentation about this point in the following text from page 21-22, line 254-292: “According to the CANMAT CPG [25], agomelatine demonstrated favorable efficacy and tolerability in a network meta-analysis of new-generation antidepressants conducted by Khoo et al (2015) [33]. The advantage of network meta-analysis is that multiple treatments can be subjected to both direct (i.e., among randomized controlled trials) and indirect (i.e., across trials based on a common variable) comparisons of interventions; the effects of different interventions that have not been investigated in trials can be compared, and the analysis of all interventions enables ranking of therapeutic alternatives with regard to a given outcome. However, other aspects must be taken into account: To assess the level of evidence or strength of the recommendation, it is important to consider not only the type of study but also its quality or risk of bias, or both, as recommended in the GRADE method, for example [29,34]. Network meta-analysis should be conducted under strict and specific methodology conditions (transitivity and consistency criteria met); the inclusion of studies at risk of bias may negatively compromise the validity of the findings [35]. In their network meta-analysis, Khoo et al did not describe the result of the risk of bias assessment for any of the eight clinical trials included in the meta-analysis, but in the general assessment, most of the primary studies included were judged as having high or unclear risk of bias in at least one domain of the Cochrane Collaboration risk of bias tool [33]. In a 2013 Cochrane review of 13 trials in a pairwise meta-analysis [36] and in a 2018 network meta-analysis conducted by Cipriani et al [3], agomelatine did not show robust advantages over the other antidepressants for effectiveness, but it appeared to be better tolerated. In both studies, the limitation was that the primary studies had biases ([3]). In addition, agomelatine was not approved by the U.S. Food and Drug Administration to treat depression, and its safety and efficacy have been questioned [36]. The Colombian CPG explicitly recommends not using agomelatine because of insufficient evidence on its effectiveness [22].

The CANMAT CPG recommendation of milnacipran as a first-line treatment is based more on its tolerability than on its efficacy [37], which does not follow the recommended sequence for the rational use of medicines: efficacy, availability, and safety [38]. Moreover, in Cipriani et al’s study [3], milnacipran did not stand out in terms of either effectiveness or safety [3]. Milnacipran is also not approved by the U.S. Food and Drug Administration to treat depression [39]. 

The recommendation of mianserin as a first-line option is controversial because the CANMAT CPG itself cites the results of a network meta-analysis that demonstrated few differences in response, although SSRIs and TCAs were superior to mianserin/mirtazapine and moclobemide [25,40]. Indeed, Linde et al [40] affirmed that physicians should be aware that SSRIs and TCAs have a somewhat more solid evidence base than do other pharmacological classes. Cipriani et al [3] did not include mianserin in network meta-analysis [3]. Arroll et al [41] conducted a meta-analysis of two trials in which mianserin was compared with placebo and concluded that mianserin was effective for continuous outcomes but did not affect rates of remission and response [41].

“Linked to the comments above, the authors, as part of their aim highlight the importance of their work for LMICs guideline developers. However, this is not demonstrated or elaborated on further in the manuscript. Do consider placing the research findings for LMICs as stated in the objectives.”

Authors’ reply: Thank you for your suggestion. Our goal was not to prove the value of this method to help LMIC countries, and the possibility of the use of our results for adaptation has already been clarified. Thus, we deleted the following phrase (which appears crossed out in version entitled “Revised Manuscript with Track Changes” [page 4, lines 77-79] from the Introduction section: “Notably, it could help to prevent redundant efforts in CPG creation, allowing low-income countries to utilize existing high-quality CPGs and thus saving valuable human and financial resources 

“The authors mention guideline adaptation, it would be useful to include what these results mean for guideline developers who would use adaptation methods (e.g. those with limited resources, not just in LMICs), what challenges it poses and what further specific research would be needed to create trustworthy CPG considering your results.”

Authors’ reply: In fact, finding high-quality guidelines increases the feasibility of adaptation projects, and by highlighting the main divergences in the recommendations, we hope to help in the planning of the adaptation process. In this way, our study will help to elaborate the CPG by means of the adaptation process as much as it clarifies important divergences between the CPGs and points that require further discussion among the stakeholders (patients and professionals) for their development. In addition, the identification of points at which CPGs converge fully or a few CPGs that have been well developed may also be helpful for the adaptation of the CPGs for local contexts and for making clinical decisions about the treatment of depression. Thus, we made the following modifications:

“Our findings could help in the elaboration/adaptation of a CPG because we identified important divergences among existing CPGs to which stakeholders should give special attention. Moreover, the identification of the points at which CPGs converge fully and that have been well addressed in certain CPGs may also be helpful for the elaboration/adaptation of a CPG for local contexts, as well as contribute to clinical decisions about treatment for this severe mental health problem”. [page 26, lines 378-384]

References

1. Colombia, Ministerio de Salud. Guía de Práctica Clínica: detección temprana y diagnóstico del episodio depresivo y trastorno depresivo recurrente en adultos: atención integral de los adultos con diagnóstico de episodio depresivo o trastorno depresivo recurrente [cited 2017 June 30; Internet]. Bogotá: Ministerio de Salud; c2013. Available from: http://gpc.minsalud.gov.co/gpc_sites/Repositorio/Conv_500/GPC_td/gpc_td.aspx.

2. Kennedy SH, Lam RW, McIntyre RS, Tourjman SV, Bhat V, Blier P, et al. Canadian Network for Mood and Anxiety Treatments (CANMAT) 2016: clinical guidelines for the management of adults with major depressive disorder: Section 3. Pharmacological Treatments. Can J Psychiatry. 2016;61: 540-560.

3. Gelenberg AJ, Freeman MP, Markowitz JC, Rosenbaum JF, Thase ME, Trivedi MH, et al. Practice guideline for the treatment of patients with major depressive disorder, third edition. [Washington, D.C.]: American Psychiatry Association; 2010.

4. Molino CGRC, Leite-Santos NC, Gabriel FC, Wainberg SK, Vasconcelos LP, Silva RAM, et al. Factors associated with high-quality guidelines: a systematic review and appraisal of 421 guidelines for the pharmacological management of chronic diseases in primary care. JAMA Intern Med. 2019;179: 553-560.

5. The ADAPTE Colaboration. The ADAPTE process: resource toolkit for guideline adaptation. Version 2.0 [Internet]. [n.p.]: The ADAPTE Colaboration; 2009. Available from: https://www.g-i-n.net/document-store/working-groups-documents/adaptation/adapte-resource-toolkit-guideline-adaptation-2-0.pdf

---

## [Decision Letter · Decision Letter 1]

31 Mar 2020

Pharmacological treatment of depression: a systematic review comparing clinical practice guideline recommendations

PONE-D-19-16309R1

Dear Dr. Gabriel,

We are pleased to inform you that your manuscript has been judged scientifically suitable for publication and will be formally accepted for publication once it complies with all outstanding technical requirements.

With kind regards,

Gabriele Fischer, MD

Academic Editor

PLOS ONE

Additional Editor Comments (optional):

Reviewers' comments:

Reviewer's Responses to Questions

**Comments to the Author**

1. If the authors have adequately addressed your comments raised in a previous round of review and you feel that this manuscript is now acceptable for publication, you may indicate that here to bypass the “Comments to the Author” section, enter your conflict of interest statement in the “Confidential to Editor” section, and submit your "Accept" recommendation.

Reviewer #1: All comments have been addressed

Reviewer #2: All comments have been addressed

2. Is the manuscript technically sound, and do the data support the conclusions?

Reviewer #1: (No Response)

Reviewer #2: Yes

3. Has the statistical analysis been performed appropriately and rigorously? 

Reviewer #1: (No Response)

Reviewer #2: Yes

4. Have the authors made all data underlying the findings in their manuscript fully available?

Reviewer #1: (No Response)

Reviewer #2: Yes

5. Is the manuscript presented in an intelligible fashion and written in standard English?

Reviewer #1: Yes

Reviewer #2: Yes

6. Review Comments to the Author

Reviewer #1: (No Response)

Reviewer #2: Thank you for clearly addressing reviewer comments. The authors have provided clear rationale for comments to both authors.

7. PLOS authors have the option to publish the peer review history of their article (what does this mean?). If published, this will include your full peer review and any attached files.

Reviewer #1: Yes: FILIPPO CARACI

Reviewer #2: Yes: Michael McCaul

---

## [Editor Report · Acceptance letter]

6 Apr 2020

PONE-D-19-16309R1 

Pharmacological treatment of depression: a systematic review comparing clinical practice guideline recommendations 

Dear Dr. Gabriel:

I am pleased to inform you that your manuscript has been deemed suitable for publication in PLOS ONE. Congratulations! Your manuscript is now with our production department. 

With kind regards,

on behalf of

Professor Gabriele Fischer 

Academic Editor

PLOS ONE